# Genetic Diversity of *Candida* spp. Isolates Colonizing Twins and Their Family Members

**DOI:** 10.3390/pathogens11121532

**Published:** 2022-12-13

**Authors:** Iwona Wojciechowska-Koszko, Paweł Kwiatkowski, Paulina Roszkowska, Barbara Krasnodębksa-Szponder, Michał Sławiński, Artur Gabrych, Stefania Giedrys-Kalemba, Barbara Dołęgowska, Edward Kowalczyk, Monika Sienkiewicz

**Affiliations:** 1Department of Diagnostic Immunology, Pomeranian Medical University in Szczecin, 70-111 Szczecin, Poland; 2Immunology Laboratory Independent, Public Clinical Hospital No. 2 in Szczecin, 70-111 Szczecin, Poland; 3Department of Laboratory Diagnostics, Public Clinical Hospital No. 2 in Szczecin, 70-111 Szczecin, Poland; 4Hospital Pharmacy, Public Clinical Hospital No. 2 in Szczecin, 70-111 Szczecin, Poland; 5Department of Medical Microbiology, Pomeranian Medical University in Szczecin, 70-111 Szczecin, Poland; 6Department of Laboratory Medicine, Pomeranian Medical University in Szczecin, 70-111 Szczecin, Poland; 7Department of Pharmacology and Toxicology, Medical University of Lodz, 90-752 Lodz, Poland; 8Department of Pharmaceutical Microbiology and Microbiological Diagnostic, Medical University of Lodz, 90-151 Lodz, Poland

**Keywords:** twins, RAPD-PCR, mycobiome, *Candida* spp.

## Abstract

A wide range of options for studying *Candida* species are available through genetic methods. Twins, particularly monozygotic ones and their families may be fitting subjects for studying those microorganisms. The question is: How specific can yeast flora be in an individual? The study aimed to analyze the strain relatedness among commensal yeasts isolated from various parts of the bodies of healthy people and to compare correlations between the genotypes of the isolates. Yeasts were isolated from 63 twins and their family members (*n* = 25) from the oral cavity, anus, interdigital space and navel. After species identification, *Candida albicans* (*n* = 139), *C. parapsilosis* (*n* = 39), *C. guilliermondii* (*n* = 25), *C. dubliniensis* (*n* = 11) and *C. krusei* (*n* = 9) isolates were analyzed using the random amplified polymorphic DNA polymerase chain reaction (RAPD-PCR) optimization method. The similarities between the strains were calculated based on the Dice (Sab) coefficient and are displayed graphically as dendrograms. Using cluster analysis, the following relatedness was distinguished: 13 genotypes and three unique (Un) patterns among *C. albicans*; 10 genotypes and four Un patterns among *C. parapsilosis*; three genotypes and one Un pattern among *C. guilliermondii* and *C. dubliniensis*; and three genotypes among *C. krusei* isolates. The presence of identical, similar or both genotypes among the strains isolated from family members shows the transmission of yeasts between ontocenoses in the same person and between individuals. The similarity between the genotypes of *C. albicans*, *C. guilliermondii*, *C. dubliniensis* and *C. krusei* was more remarkable than between the genotypes of *C. parapsilosis* in the strains isolated from ontocenoses of the same individual and their family members. The degrees of genetic similarity between *Candida* spp. strains isolated from monozygotic twins and those obtained from their relatives did not differ.

## 1. Introduction

The fungal microbiome in a healthy person undergoes continuous quantitative and qualitative changes during the various periods of life. The changes depend on the genetic factors of a host as well as environmental factors. Of all the fungi occurring in a healthy individual, the yeasts of the *Candida* genus and the yeast-like fungi of the *Malassezia* genus appear the most often. The main reservoir of yeasts in healthy people is the digestive tract, particularly the oral cavity. Fungi may also occur on the skin and colonize the urogenital system. It is worth noting that some species of *Malassezia* are also a natural part of the human skin microflora. Nevertheless, they can sometimes cause skin diseases, making them count among the opportunistic mycopathogens that cause dermatoses [1,2,3,4].

Fungi enter the human body through different pathways, such as the digestive tract, inhalation (aero-droplet and aero-powder), sexual contact, through damaged mucous membranes, cornea, or the skin [3]. They are transmitted directly from human to human and indirectly from the environment to human through contact with everyday objects [5].

Unique transmission occurs between a mother and a newborn during natural childbirth. The fungi are transmitted directly from the vagina to the oral cavity and the infant’s digestive tract and colonize the skin [6]. Phenotypic testing, which consists of assaying a biotype using Williamson’s classification and API ZYM tests, showed that the strains isolated from the mothers and their children included *C. albicans*, *C. kefyr*, *C. tropicalis*, *C. guilliermondii* and *C. krusei* with identical hydrolase profiles. 

Changes that occur in the original fungal flora, transmitted from mother to child at birth, during an individual’s life have not been explained [7]. Horizontal transmission of *Candida* strains among children and their parents and occasionally their grandparents, as well as the colonization of different body sites of the same person, have been seen. The transmission of yeasts among different body sites of the same individual was also noticed. In both cases, the transmission may have involved the oral cavity, anus, vagina, between the toes and interdigital spaces [6,8,9,10].

Molecular biology methods are now widely used to assess *Candida* spp. relatedness. They can be used not only for species identification but also for verifying intraspecies diversity. The random amplified polymorphic DNA polymerase chain reaction (RAPD-PCR) method was used in the current study. This method amplifies genomic DNA with primers of arbitrary nucleotide sequences. The method does not require pledging the details of the target DNA sequence [11]. The PCR is conducted at low annealing temperatures of 36 °C. During this reaction, each primer gives a different pattern of PCR products, and thus detecting polymorphisms among strains is possible. Statistically, the differences among strains depend on the number of analyzed bands [11].

The transmission of yeasts between twins, particularly monozygotic ones, is little studied. There are minimal data on the transmission of fungi within twins or among relatives. It is not known how twins’ zygosity affects the relatedness of the *Candida* strains, particularly in monozygotic twins with identical genetic material. Twins—particularly monozygotic ones—and their families may be fitting subjects to study these yeasts. Hence, this study aimed to analyze the strain relatedness among commensal yeasts isolated from various parts of the body of dizygotic and monozygotic twins and their family members. To compare correlations between the genotypes of the isolates, the RAPD-PCR method was used.

## 2. Materials and Methods

### 2.1. Characteristics of the Study Group

A total of 88 people (twins [*n* = 63], mothers [*n* = 15], fathers [*n* = 7] and siblings [*n* = 3]) from 33 families (F1–F33) were examined in the current study. In the case of 30 families, pairs of monozygotic (*n* = 12) and dizygotic (*n* = 18) twins were analyzed, while, in the remaining families (F2, F3 and F24), the other twin was not present or did not consent to the study. Individual family members were conventionally labeled according to the following criteria: T1—twin 1, T2—twin 2, M—mother, F—father and SIB—siblings. The average ages of T1 (*n* = 33), T2 (*n* = 30), M (*n* = 15), F (*n* = 7) and SIB (*n* = 3) were 16, 16, 37, 35 and 9 years, respectively. Details of the study group, including the distribution of family members, their age and twin type, are shown in Table 1.

### 2.2. Mycological Study

Yeasts were isolated from four body sites: oral cavity, anus, navel and interdigital space. Overall, 472 *Candida* spp. isolates were collected and analyzed for species identification using the germ tube test and biochemical ATB ID32C test (bioMérieux, Warsaw, Poland). Only strains present in at least two family members, including at least one twin, were chosen for genetic analysis. Altogether 223 isolates, including *C. albicans* (*n* = 139)*, C. parapsilosis* (*n* = 39), *C. guilliermondii* (*n* = 25), *C. dubliniensis* (*n* = 11) and *C. krusei* (*n* = 9), were analyzed. 

The relationships among the *Candida* spp. isolates and the twin zygosity were based on the analysis of 60 strains isolated from 12 pairs of monozygotic twins. Using the short tandem repeat (STR) multiplex PCR, twin zygosity was assayed at the Institute of Forensic Medicine, Wroclaw Medical University. Three tests with different numbers of assayed *loci* were used: AmpFI Cofiler, AmpFI Profiler and AmpFI SGM Plus [12,13].

### 2.3. RAPD-PCR Method

Yeast chromosomal DNA was extracted using the Graham method [14]. During the analysis, the amplification conditions followed the parameters used in the studies by Tavanti et al. [15] except for the concentrations of the PCR mixture with a final volume of 25 µL (10× PCR buffer, 2.5 mM MgCl_2_, 1.5 mM dNTP, 30 pm of each primer, 30 ng/L DNA and 1 U of Taq DNA polymerase). All reagents were bought from Roche Diagnostics GmbH (Mannheim, Germany). 

Then, using the optimized RAPD-PCR method, isolates of all *Candida* species were subjected to intraspecific analysis. The optimization of the RAPD-PCR and the choices of primers were selected individually for each *Candida* sp. isolate following the protocol suggested by Cobb and Clarkson [16]. The PCR mixture had the best constant number of reagents for each *Candida* sp. strain: 10x PCR buffer (100 mM Tris-HCl, 500 mM KCl and pH 8.3), 1 U of Taq DNA polymerase (Roche Diagnostics GmbH, Germany) and remaining components, such as MgCl_2_, dNTP and primers (Appendix A).

Additionally, to verify the species differentiation—*C. orthopsilosis, C. metapsislosis* and *C. parapsilosis*, in the case of 39 strains identified in biochemical tests as *C. parapsilosis*, additional genetic analysis was performed using the RAPD method using the RP02 primer with the following sequence 5’-GCGATCCCCA-3’ [15,17,18].

The amplification reactions were conducted on the GeneAmp PCR System 9600 thermocycler (Perkin-Elmer, Waltham, MA, USA). The RAPD-PCR conditions for all the primers used included 42 cycles. The following temperatures and times for each profile were chosen: denaturation at 94 °C for 1 min, annealing at 36 °C for 1 min and elongation at 72 °C for 1 min. The final elongation ran at 72 °C for 7 min. The amplification products, negative control and DNA molecular mass marker were analyzed using horizontal electrophoresis in 2% agarose gel stained with 0.5 mg/L ethidium bromide (Sigma-Aldrich, Poznan, Poland) in 1× TBE running buffer at 120 V for one hour. 

DNA bands were visualized under UV light and photographed. The results were assessed using BioGenProfil software (BioProfil-BioGen Windows Application Version 99.04, Glostrup, Denmark). The similarity was clustered using the Unweighted Pair Group Method with Arithmetic Mean (UPGMA) and computed based on band positions using the Dice coefficient (Equation (1)):(1)$$S_{\mathrm{AB}}=\frac{2a}{\left(2a+b+c\right)}$$
where: a—number of similar bands, b and c—numbers of different bands in two compared lanes. The S_AB_ value can range from 0 (no similar bands in two compared profiles) to 1 (all bands identical).

### 2.4. Statistical Analysis

The degree of similarity among the strains isolated from twins and the strains isolated from their relatives were compared using the Wilcoxon–Mann–Whitney test (Statistica 5.0 software version Edition, Szczecin, Poland). All values of *p* < 0.05 were considered significant.

## 3. Results

### 3.1. Source of Isolation

*Candida* spp. strains were isolated from various sources, most from the oral cavity—139 (62.3%). Strains of *C. albicans*, *C. dubliniensis* and *C. krusei* were mainly isolated from the oral cavity (55.6–73.4%), while *C. parapsilosis* and *C. guilliermondii* were isolated from interdigital space (44.0–46.2%). The frequency of isolated *Candida* spp. strains from each source is shown in Figure 1.

### 3.2. Genotypes of Candida *spp.*

The amounts of genotypes that adopted adequate S_AB_ level were ≥80% for *C. albicans* and ≥70% for the four other *Candida* species strains. The value of the S_AB_ level determined the average sum of all the S_AB_ values obtained for individual dendrograms. The genotypes obtained for each *Candida* sp. are shown in Table 2.

Thirteen genotypes (A–M) and three unique (Un) patterns were found in 139 strains of *C. albicans* isolated from 26 families (Figure 2a). Most *C. albicans* strains, isolated from different ontocenoses of the same person, belonged to the same genotype, i.e., they had identical or similar patterns, e.g., family no. 4—genotype C4; family 7—genotype D1. Only in single cases different genotypes or Un patterns were found within a family, e.g., family no. 6: T1—oral cavity: genotype D2, anus: genotype C1; family no. 9, T1—oral cavity: genotype J4, interdigital space: Un2. In a few cases, *C. albicans* strains with different genotypes were present in the same ontocenosis, e.g., family no. 6, T2—oral cavity: genotypes C1 and D2; family no. 16, T1—oral cavity: genotypes J2, F1 and F2.

After the first analysis of species diversity, 36 bands profiles characteristic of *C. parapsilosis*, two bands profiles characteristic of *C. orthopsislosis* (F12_T1_OC2_CP, F31_IS_T2_CP) and one band profile characteristic of *C. metapsilosis* (F17_T1_OC_CP) were obtained. On the other hand, the analysis of intra-species differentiation allowed for the identification of 10 genotypes (A–J), and four Un patterns were found in 39 strains of *C. parapsilosis* isolated from 10 families (Figure 2b). *C. parapsilosis* isolates from different ontocenoses of the same person or their family members showed significantly greater genetic diversity when compared with *C. albicans* strains. 

In individual cases, identical genotypes were found in different ontocenoses of the same person (family no. 26, T1—oral cavity, anus and interdigital space: genotype F1; family no. 9, T2—oral cavity and anus: genotype A2) as well as in different ontocenoses of their family members (family no. 2, T2 and M: genotype E, family no. 31, T1, T2, M, F: genotype J). Identical genotypes occurred within ontocenoses of the same person (family no. 8, T2—oral cavity: four strains belonged to genotype H1). Furthermore, diverse genotypes within Un patterns (family no. 12, T1—oral cavity: genotype B1 and Un2, family no. 15, T1—interdigital space: genotype C and Un3; M—interdigital space: genotypes C and D2) were detected.

Three genotypes (A–C) and one Un pattern were found in the 25 strains of *C. guilliermondii* isolated from seven families (Figure 2c). *C. guilliermondii* strains, such as *C. albicans* strains, showed a small genetic diversity both among and within ontocenoses of the same person and in ontocenoses of their family members. In most cases, isolates of identical or similar patterns were found. Only in family no. 19, a Un pattern—Un1 was found, in addition to similar genotypes C1 and C3, in the T1 oral cavity ontocenosis.

Three genotypes (A–C) and one Un pattern were found in 11 isolates of *C. dubliniensis* isolated from three twin pairs (Figure 2d). Identical or similar genotypes were detected in families no. 18, 19 and 20, and only one person T1 from family no. 20 was found with genotype C and the Un pattern—Un1. Three genotypes (A–C) were found in the nine *C. krusei* species isolated from three families (Figure 2d). In family no. 1 in T1, an identical genotype was found in the ontocenoses A1; in families no. 9 and 17, similar genotypes were found: B1 and B2, as well as C1, C2 and C3.

A detailed summary of the study group and the isolated *Candida* spp. strains from different ontocenoses, including their genetic relatedness obtained by RAPD-PCR, is presented in Appendix A. Representative examples of dendrograms are shown in Appendix A.

### 3.3. Genetic Analysis of Candida Isolates from Monozygotic Twins

To determine a possible correlation between twin zygosity and the degree of relationship between *Candida* strains, the genotype distribution within 60 strains was isolated from 12 pairs of monozygotic twins. The strains belonged to three species: *C. albicans*—46 strains, *C. guilliermondii*—six strains and *C. parapsilosis*—eight strains. The most remarkable genetic similarity of the strains isolated from monozygotic twins was found among *C. albicans* and *C. guilliermondii* isolates. 

Identical and similar patterns in both twins were found most often. *C. parapsilosis* strains were detected only in two pairs of monozygotic twins. One pair had similar patterns (family no. 12) and the other Un pattern (family no. 10), and the second had different patterns (family no. 17). The most monozygotic twin pair—11 (78.6%), were living together. However, the other three pairs from families no. 21, 22 and 32 did not live together.

The correlation between the genetic relatedness of strains isolated from monozygotic twins and the test results obtained from other twins and their parents were analyzed using the Wilcoxon–Mann–Whitney test. In these two groups, *C. albicans* and *C. guilliermondii* showed more remarkable similarity compared with *C. parapsilosis;* however, those differences were not statistically significant. The yeast genetic similarity coefficients among relatives, as defined by the average and standard deviations, are listed in Table 3.

## 4. Discussion

The acquisition and adaption of physiological flora is the effect of various relationships among various microorganisms and the interactions between microorganisms and the host. Genetic factors of macro-organisms play a significant role here, being responsible, among others, for the chemical composition of secretions, expression of surface receptors on mucous membranes and skin, which affect the adequate adhesion and colonization by microorganisms, including fungi. The studies based on phenotypic methods indicate the transmission of *Candida* strains between family members, in particular those who cohabit [5]. 

Yeast transmission among different ontocenoses of the same person and the ontocenoses of close relatives have been confirmed by fingerprinting methods. Genetically identical strains were found in the oral cavity and the anus in 65.2% of 23 examined preschoolers [19]. Another study on the same pairs three times within six-month intervals also showed interpersonal transmission among relatives in a mother—newborn relationship. 

The results suggest that colonization with fungal flora, which follows birth, continues through the individual’s life. This flora is derived mainly from the mother and the surroundings. Another group of children was colonized by strains different from their mothers’ [20]. This confirms the possibility of colonization of human ontocenoses with strains originating from exogenous sources. Similarly, molecular studies by Abdeljelil et al. [21] conducted in hospital conditions confirm the transmission of *C. albicans* strains between medical staff and newborns in the intensive care unit.

Our research analyzed the degree of diversity/relatedness of *Candida* strains isolated from twins and their family members. High genetic similarities were found among *Candida* strains isolated from various ontocenoses of twins and their parents. In most families, identical and similar genetic patterns were found. This shows that the transmission of the same yeast clones among ontocenoses of a given person and the transmission of the same strains among family members, are likely to occur.

The occurrence of genetically similar isolates in different ontocenoses of the same person or different ontocenoses of their family members may be caused by allelism and chromosome reassortments occurring during the parasexual reproductive cycle of yeasts [22]. These changes may also be associated with different environmental factors or caused by other genetic or immunological determinants. A slight change in the genetic profile of the strain (similar type) may be related to the adaptation of the strain to different conditions in a new ontocenosis, such as a transmission from the oral cavity to the anus or the vagina. 

Genetic tests that analyzed *C. albicans* strains isolated from the oral cavity and the anus of people suffering from Crohn’s disease and members of their families using the multilocus sequence typing (MLST) method seem to confirm the theory of the inter-ontocenosis and interfamilial transmission of fungi [22]. These results also showed identical or closely related strains within the examined ontocenoses in the studied group of patients and their relatives.

The occurrence of strains with different genotype patterns, or unique patterns within families, might affect the acquisition of a new strain from the environment, showing a periodic exchange of yeast strains colonizing various human ontocenoses. It is worth noting that, in the current study, the most remarkable genetic diversity was found in the strains of *C. parapsilosis*, which, during the analysis, showed the most substantial number of unique bands—four. 

In the case of these isolates, preliminary molecular analysis regarding species diversity showed that two of them belong to *C. orthopsilosis* species (F12_T1_OC2_CP, F31_IS_T2_CP) and the rest—one to *C. parapsilosis* (F15_T1_IS2_CP) and the other to *C. metapsilosis* (strain 1-F17_T1_OC_CP). The genetic diversity among *C. parapsilosis* isolates was also confirmed by Delfino et al. [23], who showed that *C. parapsilosis* strains isolated from the hands of medical personnel were significantly more heterogeneous than clinical strains.

Monozygotic twins seem particularly fitting subjects for studying factors mediating yeast colonization. In our studies, *C. albicans, C. parapsilosis* and *C. guilliermondii* strains isolated from 24 monozygotic twins (12 pairs) were tested for genetic relatedness and then compared to the isolates from the group consisting of dizygotic twins and their parents. In both groups, the similarity ratios were high. The similarity between the group of *C. albicans* and *C. guilliermondii* isolates from monozygotic twins and the isolates from the remaining group of twins and their parents was higher than between *C. parapsilosis* strains. 

A higher diversity of genetic types predominating similar and unique patterns was found within the *C. parapsilosis* isolates from family members. However, there was no distinct correlation between the fungi isolates from monozygotic twins and other members of their families. It is thus possible that, in people that are genetically closely related—for example, monozygotic twins—there is no greater predisposition for colonization with genetically identical *Candida* strains, irrespective of their cohabitation. On the other hand, since the group of the studied subjects was too small to stand for the general population, these conclusions should be treated as preliminary results.

Another observation in our study was that *Candida* strains isolated from and within a particular family belonged, in general, to various genotypes. Only in a few families were the same or similar genotypes found. In addition, the genetic variability of *Candida* strains isolated from inhabitants of geographically different regions was observed [24]. This confirms a significant diversity of *Candida* strains circulating in the environment and shows their adaptability to labile/variable environmental conditions when shifting from one ontocenosis to another.

We found that the degree of relatedness of *Candida* strains in the group of monozygotic twins and their family members was the same as in other related people. A higher genetic similarity was found among *C. albicans* and *C. guilliermondii* than was found among *C. parapsilosis* strains.

## 5. Conclusions

We concluded that higher genetic similarity of strains was found within *C. albicans* and *C. guilliermondii* compared with within *C. parapsilosis* isolates. The degrees of relatedness among *Candida* spp. isolates from the monozygotic twins were similar and were statistically similar to other related people. The convergence of the results obtained with RAPD and MLST confirms (despite certain limitations and reservations) the usefulness of the first method in epidemiological examinations, and there is also an economic advantage.

## Figures and Tables

**Figure 1 pathogens-11-01532-f001:**
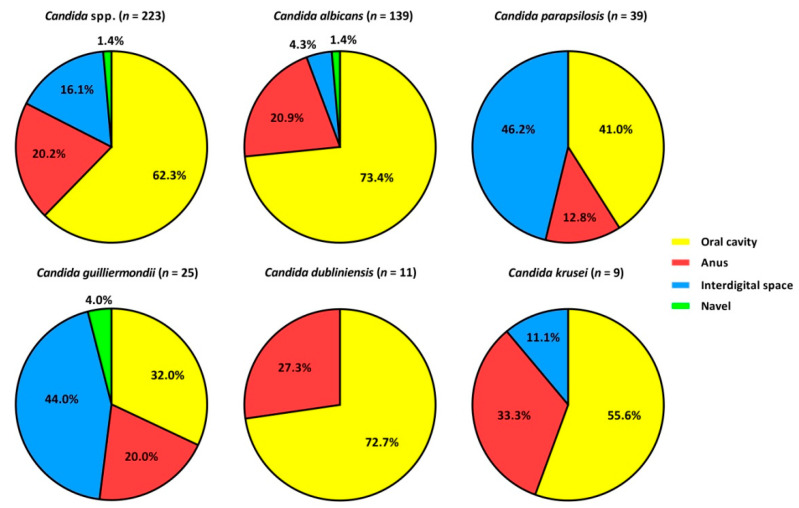
The frequencies of the isolated *Candida* spp. strains.

**Figure 2 pathogens-11-01532-f002:**
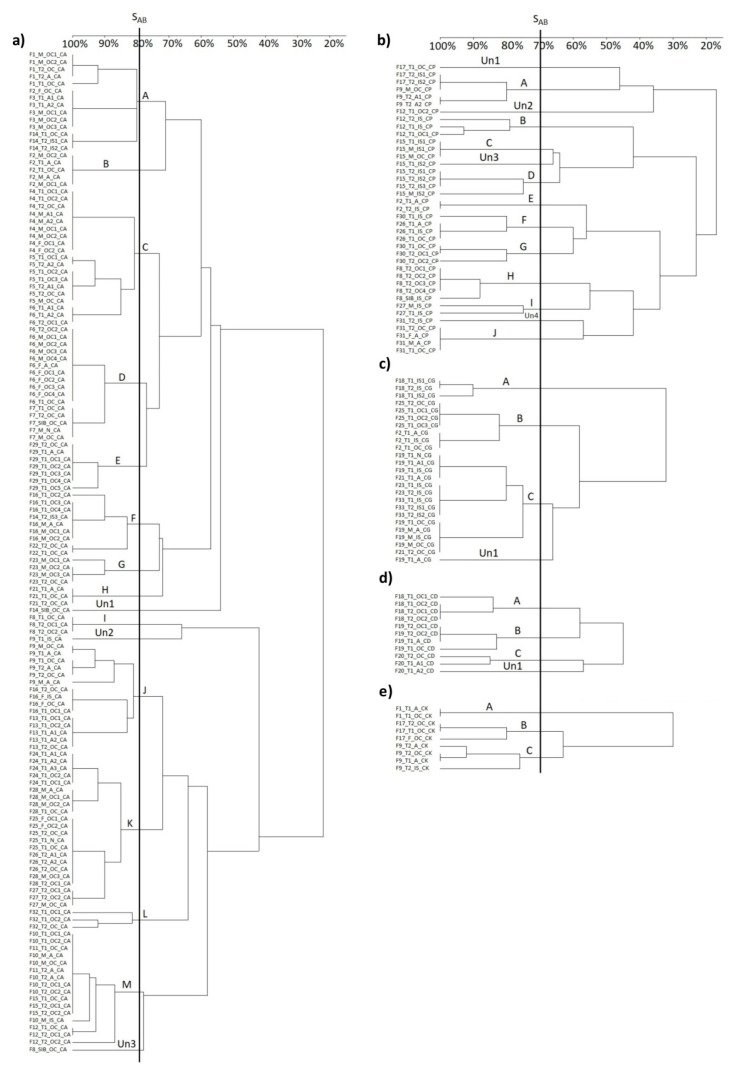
Dendrograms of: (**a**) *Candida albicans*, (**b**) *C. parapsilosis*, (**c**) *C. guilliermondii*, (**d**) *C. dubliniensis* and (**e**) *C. krusei* isolates. Strains described according to the following pattern: number of family_member of family_source of isolation_species.

**Table 1 pathogens-11-01532-t001:** Characteristics of the study group by a distribution of family members, age and twin type.

No.	No. of Family (F)	Twin Type	Family Member (Age in Years)				
			Twin 1	Twin 2	Mother	Father	Siblings
1.	F1	Dizygotic	+ (12)	+ (12)	+ (42)	-	-
2.	F2	-	+ (6)	-	-	+ (32)	-
3.	F3	-	+ (11)	-	+ (46)	-	-
4.	F4	Monozygotic	+ (14)	+ (14)	+ (43)	+ (45)	-
5.	F5	Monozygotic	+ (14)	+ (14)	+ (40)	-	-
6.	F6	Monozygotic	+ (2)	+ (2)	+ (29)	+ (29)	-
7.	F7	Dizygotic	+ (12)	+ (12)	+ (36)	-	+ (10)
8.	F8	Monozygotic	+ (9)	+ (9)	-	-	+ (11)
9.	F9	Dizygotic	+ (1)	+ (1)	+ (34)	-	-
10.	F10	Monozygotic	+ (13)	+ (13)	+ (41)	-	-
11.	F11	Dizygotic	+ (21)	+ (21)	-	-	-
12.	F12	Monozygotic	+ (15)	+ (15)	-	-	-
13.	F13	Monozygotic	+ (17)	+ (17)	-	-	-
14.	F14	Dizygotic	+ (2)	+ (2)	-	-	+ (6)
15.	F15	Dizygotic	+ (9)	+ (9)	+ (39)	-	-
16.	F16	Monozygotic	+ (11)	+ (11)	+ (34)	+ (39)	-
17.	F17	Dizygotic	+ (13)	+ (13)	-	+ (35)	-
18.	F18	Dizygotic	+ (27)	+ (27)	-	-	-
19.	F19	Dizygotic	+ (11)	+ (11)	+ (47)	-	-
20.	F20	Dizygotic	+ (50)	+ (50)	-	-	-
21.	F21	Monozygotic	+ (67)	+ (67)	-	-	-
22.	F22	Monozygotic	+ (34)	+ (34)	-	-	-
23.	F23	Dizygotic	+ (8)	+ (8)	+ (36)	-	-
24.	F24	-	+ (26)	-	-	-	-
25.	F25	Monozygotic	+ (2)	+ (2)	-	+ (34)	-
26.	F26	Dizygotic	+ (24)	+ (24)	-	-	-
27.	F27	Dizygotic	+ (1)	+ (1)	+ (27)	-	-
28.	F28	Dizygotic	+ (17)	+ (17)	+ (40)	-	-
29.	F29	Dizygotic	+ (15)	+ (15)	-	-	-
30.	F30	Dizygotic	+ (13)	+ (13)	-	-	-
31.	F31	Dizygotic	+ (2)	+ (2)	+ (27)	+ (29)	-
32.	F32	Monozygotic	+ (44)	+ (44)	-	-	-
33.	F33	Dizygotic	+ (9)	+ (9)	-	-	-

**Table 2 pathogens-11-01532-t002:** Genotypes of *Candida* spp. obtained from twins and their relatives.

No. of Family	Genotypes of *Candida* spp. ^No. of Isolates^																			
	Twin 1				Twin 2				Mother				Father				Siblings			
	OC	AN	IS	NA	OC	AN	IS	NA	OC	AN	IS	NA	OC	AN	IS	NA	OC	AN	IS	NA
F1	A3^1^	A1^1^	-	-	A4^1^	A4^1^	-	-	A4^2^	-	-	-	-	-	-	-	-	-	-	-
	A1^1^																			
F2	-	-	-	-	B^1^	B^1^	B2^1^	-	B^2^	B^1^	E^1^	-	A2^1^	-	-	-	-	-	-	-
						E^1^														
					B2^1^	B2^1^														
F3	-	A2^2^	-	-	-	-	-	-	A2^3^	-	-	-	-	-	-	-	-	-	-	-
F4 *	C4^2^	-	-	-	C4^1^	-	-	-	C4^2^	C4^2^	-	-	C4^2^	-	-	-	C4^1^	-	-	-
F5 *	C2^2^ C3^1^	-	-	-	C2^1^	C2^1^ C3^1^	-	-	C2^1^	-	-	-	-	-	-	-	-	-	-	-
F6 *	D2^1^	C1^2^	-	-	C1^1^ D2^1^	-	-	-	D2^4^	-	-	-	D2^4^	D2^1^	-	-	-	-	-	-
F7	D1^1^	-	-	-	D1^1^	-	-	-	D1^1^	-	-	D1^1^	-	-	-	-	D1^1^	-	-	-
F8 *	I^1^	-	-	-	I^2^	-	-	-	-	-	-	-	-	-	-	-	Un3^1^	-	H2^1^	-
					H1^4^															
F9	J4^1^	J5^1^	Un2^1^	-	J4^1^	J4^1^	C3^1^	-	J5^1^	J3^1^	-	-	-	-	-	-	-	-	-	-
					A2^1^	A2^1^			A1^1^											
		C2^1^			C2^1^	C1^1^														
F10 *	M3^2^	-	-	-	M3^2^	M3^1^	-	-	M3^1^	M3^1^	M4^1^	-	-		-	-	-	-	-	-
F11	M3^1^	-	-	-	-	M3^1^	-	-	-	-	-	-	-	-	-	-	-	-	-	-
F12 *	M2^1^	-	B2^1^	-	M1^1^ M2^1^	-	B3^1^	-	-	-	-	-	-	-	-	-	-	-	-	-
	B1^1^ Un2^1^																			
F13 *	J1^2^	J1^2^	-	-	J1^1^	-	-	-	-	-	-	-	-	-	-	-	-	-	-	-
F14	A1^1^	-	-	-	-	-	A1^2^ F1^1^	-	-	-	-	-	-	-	-	-	Un1^1^	-	-	-
F15	M3^1^	-	C^1^ Un3^1^	-	M3^2^	-	D1^3^	-	C^1^	-	C^1^ D2^1^	-	-	-	-	-	-	-	-	-
F16 *	J2^1^ F2^2^ F1^1^	-	-	-	J2^1^	-	-	-	F2^2^	F2^1^	-	-	J2^1^	-	J2^1^	-	-	-	-	-
F17	Un1^1^	-	-	-	B1^1^	-	A1^2^	-	-	-	-	-	B2^1^	-	-	-	-	-	-	-
	B1^1^																			
F18	A1^1^ A2^1^	-	A1^1^ A2^1^	-	A1^2^	-	A1^1^	-	-	-	-	-	-	-	-	-	-	-	-	-
F19	C3^1^	C1^1^ Un1^1^	C1^1^	C1^1^	B2^2^	-	-	-	C3^1^	C3^1^	C3^1^	-	-	-	-	-	-	-	-	-
	B1^1^	B2^1^																		
F20	-	C1^1^ Un1^1^	-	-	C2^1^	-	-	-	-	-	-	-	-	-	-	-	-	-	-	-
F21 *	H^1^	H^1^	-	-	H^1^	-	-	-	-	-	-	-	-	-	-	-	-	-	-	-
		C1^1^			C3^1^															
F22 *	F3^1^	-	-	-	F3^1^	-	-	-	-	-	-	-	-	-	-	-	-	-	-	-
F23	-	-	C2^1^	-	G1^1^	-	C2^1^	-	G1^2^ G2^1^	-	-	-	-	-	-	-	-	-	-	-
F24	K4^2^	K4^3^	-	-	-	-	-	-	-	-	-	-	-	-	-	-	-	-	-	-
F25 *	K2^1^	-	-	K2^1^	K2^1^	-	-	-	-	-	-	-	K2^2^	-	-	-	-	-	-	-
	B1^3^				B1^1^															
F26	F1^1^	F1^1^	F1^1^	-	K2^1^	K2^2^	-	-	-	-	-	-	-	-	-	-	-	-	-	-
F27	-	-	I1^1^	-	K1^2^	-	-	-	K1^1^	-	I2^1^	-	-	-		-	-	-	-	-
F28	K3^1^	-	-	-	K2^1^	-	-	-	K2^1^ K3^2^	K3^1^	-	-	-	-	-	-	-	-	-	-
F29	E2^4^ E1^1^	E2^1^	-	-	E2^1^	-	-	-	-	-	-	-	-	-	-	-	-	-	-	-
F30	G1^1^	-	F2^1^	-	G1^1^ G2^1^	-	-	-	-	-	-	-	-	-	-	-	-	-	-	-
F31	J^1^	-	-	-	J^1^	-	Un4^1^	-	-	J^1^	-	-	-	J^1^	-	-	-	-	-	-
F32 *	L2^1^ L3^1^	-	-	-	L1^1^	-	-	-	-	-	-	-	-	-	-	-	-	-	-	-
F33	-	-	C2^1^	-	-	-	C2^2^	-	-	-	-	-	-	-	-	-	-	-	-	-

OC—oral cavity; AN—anus; IS—interdigital space; and NA—navel. The genotypes of the different *Candida* strains are marked with the following colors: yellow—*Candida albicans*, blue—*Candida parapsilosis*, green—*Candida guilliermondii*, red—*Candida dubliniensis* and gray—*Candida krusei*; and *—family in which affirmed occurrence monozygotic twins.

**Table 3 pathogens-11-01532-t003:** Statistical analysis of *Candida* species relatedness—the arithmetical average of coefficient S_AB_ ± standard deviation.

*Candida* spp.	Twins	Combination					
		T1–T2	T1–M	T1–F	T2–M	T2–F	M–F
*C. albicans*	M	0.892 ± 0.056	0.926 ± 0.121	0.769 ± 0.298	0.874 ± 0.173	0.927 ± 0.000	0.870 ± 0.298
*C. parapsilosis*		0.568 ± 0.226	-	-	-	-	-
*C. guilliermondii*		0.958 ± 0.176	-	-	-	-	-
*C. albicans*	D	0.867 ± 0.211	0.921 ± 0.128	0.789 ± 0.301	0.883 ± 0.188	0.898 ± 0.133	0.870 ± 0.298
*C. parapsilosis*		0.66 ± 0.148	0.796 ± 0.000	1 *	0.715 ± 0.141	0.790 ± 0.296	1 *
*C. guilliermondii*		0.931 ± 0.114	0.782 ± 0.176	-	-	-	-

M—monozygotic twins; D—dizygotic twins; T1—twin 1; T2—twin 2; M—mother; and F—father; * in these cases, only one pair of isolates was present among comparing ontocenoses; thus, there is no standard deviation.

## Data Availability

The data are contained within the article.

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
