# Peer review of "Genetic Diversity of Candida spp. Isolates Colonizing Twins and Their Family Members"

_pathogens, 2022, doi:10.3390/pathogens11121532_

Round 1

Reviewer 1 Report

Dear Authors,

in my opinion your work is very interesting in a cognitive context and contributes a lot to mycology, candidiasis epidemiology, molecular identification of fungi and genotyping and in getting to know the human mycobiome.

All the tables and figures are appropriate for this type of articles. In general, the paper has a logical flow, and it is refined in detail. The abstract well correspond with the main aspects of the work. Nevertheless, I see a few weak points in this work (given below), which I am convinced that the Authors are able to resolve very fast.

In my humble opinion, a serious shortcoming that I notice in this work is the lack of any information on the yeast-like fungi of the genus Malassezia, which, after all, constitute the most important component of the skin mycobiome of every human being. There is a wealth of information in the mycological literature on this subject. I would like to recommend extending the "Introduction" chapter and supplementing the information on the role of lipophilic yeast-like fungi of the genus Malassezia as the most important components of skin mycobiome.

Moreover, in my opinion, it is worth changing the discussion a bit so as not to suggest that the results obtained for other family members (mothers, n = 15, fathers, n = 7 and siblings, n = 3) are statistically significant. I believe that the results for monozygotic twins and dizygotic twins may be statistically significant, but for other family members it is better to treat it with caution.

As a reviewer I am obligated to pay attention even to less important weak points of this work and all mentioned below comments should be carefully considered.

Lines 32-33

To the best of my knowledge should be ,,isolates were analyzed using the RAPD-PCR optimized method” instead of ,,isolates was analyzed using the RAPD-PCR optimization method”

Lines 49-50

I cannot agree with the statement (quote) ,,Of all the fungi occurring in a healthy individual, the yeasts of the Candida genus appear the most frequently.” What about yeast-like fungi of the genus Malassezia ?

Lines 70-71

In my opinion, this sentence requires editing to be understandable to potential Readers.

Line 77

,,…so, this way of detecting polymorphism…” sounds better and correctly

Line 81

I would like to recommend small correction, namely ,,There are very limited data…” instead of ,,There is very little data…”

Line 82

As I know there should be ,,…the relatedness of the Candida strains…”

Lines 88-91

This sentence is too long and seems to be endless, without a conclusion ,,Hence, the study aimed to analyze strain relatedness among commensal yeasts isolated from different parts of the body of dizygotic and monozygotic twins and their family members and to compare correlations between the genotypes of the isolates using the RAPD-PCR method.”

Line 94

In my opinion ,, A total of 88 people” or ,,A total of 88 volunteers” sounds better than ,,subjects”

Line 103, Table 1

There is something wrong with a horizontal inner edge of the table 1 (between numbers 1 and 2).

Line 31, 106 and throughout the text of the manuscript

In my humble opinion ,,interdigital space” sounds really better than ,,intertoe space”

Line 108

In materials and methods, Authors declare that in addition to the species identification based on molecular methods, they conducted germ tube tests and biochemical ATB ID32C tests. Apart from that one sentence, nowhere else can you find information about the results of these experiments. It would be good if Authors included at least a table in which the results of these tests would be presented for individual strains (genotypes). Such a table could be included in the ,,Supplementary materials”.

Line 115

To the best of my knowledge proper name is ,,Wroclaw Medical University” (formerly Medical Academy of Wroclaw)

Lines 114-115

As declared by Authors of this manuscript (I quote) ,,Using the STR multiplex PCR, twin zygosity was assayed at the Institute of Forensic Medicine, Medical Academy of Wroclaw.”, at least in acknowledgments should mention and thank the person(s) from the Institute of Forensic Science for carrying out these analyzes, the results of which, as I understand, are included in the manuscript.

Lines 166-169

Italic is not used everywhere where it is necessary in the manuscript text for the species and generic names of fungi, for example page 5 and lines 166-169

Lines 197-199

Before parenthesis ,,were found" and after parenthesis ,,were detected" - it sounds like a repetition

Line 248

,,…among other things…” In my opinion ,,thing” is not good term in this case, let`s try to replace it

Line 324

To the best of my knowledge should be ,,… C. guilliermondii than within C. parapsilosis isolates” instead of ,,…C. guilliermondii then within C. parapsilosis isolates…”

Line 337

In my opinion there should be:

,,All authors have read and agreed to the final version of the manuscript.” instead of ,,published”

Author Response

Dear Reviewer 1,

We would like to thank You very much for the valuable comments concerning our manuscript: Genetic Diversity of Candida spp. Isolates Colonizing Twins and Their Family Members”. All Your suggestions were strongly taken into consideration what surely has improved manuscript’s scientific value for what we would like to express our gratitude. The changes being result of answers to Your remarks are highlighted yellow.

Dear Authors, in my opinion your work is very interesting in a cognitive context and contributes a lot to mycology, candidiasis epidemiology, molecular identification of fungi and genotyping and in getting to know the human mycobiome.

Answer: Thank you very much.

All the tables and figures are appropriate for this type of articles. In general, the paper has a logical flow, and it is refined in detail. The abstract well correspond with the main aspects of the work. Nevertheless, I see a few weak points in this work (given below), which I am convinced that the Authors are able to resolve very fast.

In my humble opinion, a serious shortcoming that I notice in this work is the lack of any information on the yeast-like fungi of the genus Malassezia, which, after all, constitute the most important component of the skin mycobiome of every human being. There is a wealth of information in the mycological literature on this subject. I would like to recommend extending the "Introduction" chapter and supplementing the information on the role of lipophilic yeast-like fungi of the genus Malassezia as the most important components of skin mycobiome.

Answer: The information have been added to Introduction section.

Moreover, in my opinion, it is worth changing the discussion a bit so as not to suggest that the results obtained for other family members (mothers, n = 15, fathers, n = 7 and siblings, n = 3) are statistically significant. I believe that the results for monozygotic twins and dizygotic twins may be statistically significant, but for other family members it is better to treat it with caution.

Answer: Thank you very much for this comment. Nevertheless, we believe that the aim of the study was to analyze genotypic Candida strains in twins and their family members. Therefore, we cannot omit this study group (other family members) from the analysis of the results and discussion. The statistical test that was used was adequate for the study group. If we exclude this study group, both the purpose and title of the paper would have to change.

As a reviewer I am obligated to pay attention even to less important weak points of this work and all mentioned below comments should be carefully considered.

Lines 32-33

To the best of my knowledge should be ,,isolates were analyzed using the RAPD-PCR optimized method” instead of ,,isolates was analyzed using the RAPD-PCR optimization method”

Answer: Done.

Lines 49-50

I cannot agree with the statement (quote) ,,Of all the fungi occurring in a healthy individual, the yeasts of the Candida genus appear the most frequently.” What about yeast-like fungi of the genus Malassezia?

Answer: Corrected.

Lines 70-71

In my opinion, this sentence requires editing to be understandable to potential Readers.

Answer: Done.

Line 77

,,…so, this way of detecting polymorphism…” sounds better and correctly

Answer: Done.

Line 81

I would like to recommend small correction, namely ,,There are very limited data…” instead of ,,There is very little data…”

Answer: Initially, we changed the sentence, but the Native Speaker decided that ‘minimal data’ fits best here.

Line 82

As I know there should be ,,…the relatedness of the Candida strains…”

Answer: Done.

Lines 88-91

This sentence is too long and seems to be endless, without a conclusion ,,Hence, the study aimed to analyze strain relatedness among commensal yeasts isolated from different parts of the body of dizygotic and monozygotic twins and their family members and to compare correlations between the genotypes of the isolates using the RAPD-PCR method.”

Answer: Done.

Line 94

In my opinion ,, A total of 88 people” or ,,A total of 88 volunteers” sounds better than ,,subjects”

Answer: Done.

Line 103, Table 1

There is something wrong with a horizontal inner edge of the table 1 (between numbers 1 and 2).

Answer: Done.

Line 31, 106 and throughout the text of the manuscript

In my humble opinion ,,interdigital space” sounds really better than ,,intertoe space”

Answer: Done.

Line 108

In materials and methods, Authors declare that in addition to the species identification based on molecular methods, they conducted germ tube tests and biochemical ATB ID32C tests. Apart from that one sentence, nowhere else can you find information about the results of these experiments. It would be good if Authors included at least a table in which the results of these tests would be presented for individual strains (genotypes). Such a table could be included in the ,,Supplementary materials”.

Answer: Done.

Line 115

To the best of my knowledge proper name is ,,Wroclaw Medical University” (formerly Medical Academy of Wroclaw)

Answer: Done.

Lines 114-115

As declared by Authors of this manuscript (I quote) ,,Using the STR multiplex PCR, twin zygosity was assayed at the Institute of Forensic Medicine, Medical Academy of Wroclaw.”, at least in acknowledgments should mention and thank the person(s) from the Institute of Forensic Science for carrying out these analyzes, the results of which, as I understand, are included in the manuscript.

Answer: A gratuity was given for the tests performed. Unfortunately, we are unable to answer who exactly was responsible for conducting them.

Lines 166-169

Italic is not used everywhere where it is necessary in the manuscript text for the species and generic names of fungi, for example page 5 and lines 166-169

Answer: Done.

Lines 197-199

Before parenthesis ,,were found" and after parenthesis ,,were detected" - it sounds like a repetition

Answer: Done.

Line 248

,,…among other things…” In my opinion ,,thing” is not good term in this case, let`s try to replace it

Answer: Initially, we changed the words, but the Native Speaker decided that ‘among others’ fits best here.

Line 324

To the best of my knowledge should be ,,… C. guilliermondii than within C. parapsilosis isolates” instead of ,,…C. guilliermondii then within C. parapsilosis isolates…”

Answer: Done.

Line 337

In my opinion there should be:

,,All authors have read and agreed to the final version of the manuscript.” instead of ,,published”

Answer: Initially, we changed the words, but the Native Speaker decided that ‘definitive version’ fits best here.

Additionally, the whole manuscript was edited by Native Speaker.

Thank You very much for Your valuable suggestion.

Sincerely Yours,

Authors

Reviewer 2 Report

Minor Issues:

Acronyms used without being defined for instance RAPD, STR,

Intro

Line 62 hydrolases should be hydrolase

Line 75 PCR reaction should be PCR

Line 81 No evidence for why this is particularly interesting is given, only that it is not known.

Line 84 It is unclear to this reviewer what is meant by the activity of their genes begins to function in a slightly different way.

Line 85 The rationalization of this is not clear, why would changes to twins genetic material lead to variety of adhesin receptors?

Line 87 what are “those”

Line 87 The question is: How specific can yeast flora be in an individual?  Does the author mean to an individual also I don’t know if specific is the right term here.I think it is known that microbiomes are specific for indivisduals  In addition, what is the hypothesis that is being tested that twins don’t have a specific microbiome?    

M and M

Line 94 n=63 is this sets of twins, just says twins.

Line 120 PCR reaction should be PCR (other places where this occurs as well and should be changed)

Line 149 no typical bands, should be no similar bands

Figure 1 is blurry when I look at it, looks like it is not 300 DPI.

3.2 I don’t know what Genetic Types of Candida species means? 

Significant Issues

Overall I think that this is a good paper but the analysis has to be more clearly explained. It is unclear if these results are repeatable or if replicates were done of the PCRs.   Since no primary data or examples are given, it is hard to determine if the conclusions are accurately drawn. I would feel more comfortable if we were given an example of what the gels looked like and how they were quantitated at least in the supplemental materials. Also if these data were repeatable through biological replicates. Also if a band had decrease in intensity was it still counted, was their a cutoff for this? How this quantitation was done will greatly increase confidence in the work.  

There was no discussion on the potential of loss of heterozygosity mutants.  

In Methods section it is unclear how the dendrogram were made, “generated using the Unweighted Pair Group Method with Arithmetic Mean” but is this of the gel pics or the SAB stats. 

Author Response

Dear Reviewer 2,

We would like to thank You very much for the valuable comments concerning our manuscript: Genetic Diversity of Candida spp. Isolates Colonizing Twins and Their Family Members”. All Your suggestions were strongly taken into consideration what surely has improved manuscript’s scientific value for what we would like to express our gratitude. The changes being result of answers to Your remarks are highlighted green.

Minor Issues:

Acronyms used without being defined for instance RAPD, STR,

Answer: Done.

Intro

Line 62 hydrolases should be hydrolase

Answer: Done.

Line 75 PCR reaction should be PCR

Answer: Done.

Line 81 No evidence for why this is particularly interesting is given, only that it is not known.

Answer: Corrected.

Line 84 It is unclear to this reviewer what is meant by the activity of their genes begins to function in a slightly different way.

Answer: The Reviewer is right, so we decided to remove this sentence.

Line 85 The rationalization of this is not clear, why would changes to twins genetic material lead to variety of adhesin receptors?

Answer: The Reviewer is right, so we decided to remove this sentence.

Line 87 what are “those”

Answer: Corrected.

Line 87 The question is: How specific can yeast flora be in an individual?  Does the author mean to an individual also I don’t know if specific is the right term here. I think it is known that microbiomes are specific for individuals  In addition, what is the hypothesis that is being tested that twins don’t have a specific microbiome?

Answer: The Reviewer is right, so we decided to remove this question. Nevertheless, we made the following hypothesis: A genetic variation within Candida strains is lower in monozygotic twins than dizygotic twins.

M and M

Line 94 n=63 is this sets of twins, just says twins.

Answer: With all due respect to the Reviewer, we don't know how to address this comment. The paper examined 63 twins. The data does not refer to pairs of twins, but to individuals. Nevertheless, if the wording twins does not fit here, we ask the Reviewer to help.

Line 120 PCR reaction should be PCR (other places where this occurs as well and should be changed)

Answer: Done.

Line 149 no typical bands, should be no similar bands

Answer: Done.

Figure 1 is blurry when I look at it, looks like it is not 300 DPI.

Answer: Our figures were generated with the quality of 600 dpi. We believe that it is presented in lower quality in the pdf version of the manuscript.

3.2 I don’t know what Genetic Types of Candida species means?

Answer: Corrected.

Significant Issues

Overall I think that this is a good paper but the analysis has to be more clearly explained. It is unclear if these results are repeatable or if replicates were done of the PCRs.   Since no primary data or examples are given, it is hard to determine if the conclusions are accurately drawn. I would feel more comfortable if we were given an example of what the gels looked like and how they were quantitated at least in the supplemental materials. Also if these data were repeatable through biological replicates. Also if a band had decrease in intensity was it still counted, was their a cutoff for this? How this quantitation was done will greatly increase confidence in the work.  

There was no discussion on the potential of loss of heterozygosity mutants.  

Answer: As suggested by the Reviewer, examples of the genotypes obtained for each Candida species are included in Supplementary Materials. The RAPD-PCR method used in the study was optimized for each species according to the scheme proposed by Taguchi and Wu and modified by Cobb and Clarkson. Additionally, repeatability was checked for selected results. Bands that were faintly visible were also included in the analysis. Unfortunately, the potential of loss of heterozygosity mutants was not included in the study. This work is currently being carried out in our laboratory.

Taguchi G, Wu Y. Introduction to Off-Line Quality Control. Japan Quality Control Organisation, Nagoya 1980.

Cobb BD. Optimalization of RAPD fingerprinting. [in:] Micheli MP, Bova R, eds. Fingerprinting methods based on arbitrarily primed PCR. Berlin, New York: Springer-Verlag; 1997. p.29-34.

In Methods section it is unclear how the dendrogram were made, “generated using the Unweighted Pair Group Method with Arithmetic Mean” but is this of the gel pics or the SAB stats.

Answer: The similarity of RAPD-profiles for Candida spp. was calculated using the Dice similarity coefficient (SAB), clustered by the Unweighted Pair Group Method with Arithmetic Mean (UPGMA) and visualized by the dendrogram.

Additionally, the whole manuscript was edited by Native Speaker.

Thank You very much for Your valuable suggestions.

Sincerely Yours,

Authors